# Protective Effect of Escitalopram on Hepatocellular Carcinoma by Inducing Autophagy

**DOI:** 10.3390/ijms23169247

**Published:** 2022-08-17

**Authors:** Li-Jeng Chen, Tsai-Ching Hsu, Hsiang-Lin Chan, Chiao-Fan Lin, Jing-Yu Huang, Robert Stewart, Bor-Show Tzang, Vincent Chin-Hung Chen

**Affiliations:** 1Institute of Medicine, Chung Shan Medical University, Taichung 40201, Taiwan; 2Immunology Center, Chung Shan Medical University, Taichung 40201, Taiwan; 3Department of Clinical Laboratory, Chung Shan Medical University Hospital, Taichung 40201, Taiwan; 4Department of Child Psychiatry, Linkou Chang Gung Memorial Hospital, Taoyuan 33305, Taiwan; 5Department of Psychiatry, School of Medicine, Chang Gung University, Taoyuan 33302, Taiwan; 6Department of Psychiatry, Chang Gung Medical Foundation, Chiayi Chang Gung Memorial Hospital, Chiayi 61303, Taiwan; 7Institute of Psychiatry, Psychology and Neuroscience, King’s College London, London WC2R 2LS, UK; 8South London and Maudsley NHS Foundation Trust, London SE5 8AZ, UK; 9Department of Biochemistry, School of Medicine, Chung Shan Medical University, Taichung 40201, Taiwan

**Keywords:** hepatocellular carcinoma (HCC), selective serotonin reuptake inhibitors (SSRIs), escitalopram, autophagy, nationwide population-based cohort study

## Abstract

Background: Hepatocellular carcinoma (HCC) is an aggressive cancer with poor prognosis. Although recent research has indicated that selective serotonin reuptake inhibitors (SSRIs), including escitalopram, have anticancer effects, little is known about the effects of escitalopram on HCC. Methods: Both in vitro and in vivo studies were conducted to verify the potentials of escitalopram on HCC treatment. To explore whether the effects of escitalopram are clinically consistent with laboratory findings, a nationwide population-based cohort study was also adopted to examine the association between escitalopram and HCC risk. Results: As compared with THLE-3 cells, escitalopram significantly inhibited the proliferation of HepG2 and Huh-7 cells. Specifically, escitalopram significantly induced autophagy in HepG2 and Huh-7 cells by increasing the LC3-II/LC3-I ratio and the expression of ATG-3, ATG-5, ATG-7, and Beclin-1 proteins. Moreover, escitalopram significantly inhibited the growth of xenografted Huh-7 cells in SCID mice that were treated with 12.5 mg/kg escitalopram. Accordingly, the risk of HCC was negatively correlated with escitalopram use. Conclusions: These findings provided evidence supporting the therapeutic potential of escitalopram for HCC. Both laboratory and nationwide population-based cohort evidence demonstrated the attenuated effects of escitalopram on HCC.

## 1. Introduction

A recent systematic analysis indicated that nearly 534,000 incident cases of liver cancer were diagnosed in 2019, and nearly 485,000 deaths due to liver cancer occurred globally [1]. Currently, liver cancer is the third leading cause of cancer death worldwide [2,3]. Hepatocellular carcinoma (HCC) accounts for 75–85% of liver cancers [2]. In the United States, the HCC 2-year survival rate is less than 50%, and the 5-year survival rate is only 10%, indicating poor prognosis [4]. HCC patients often have psychological comorbidity, particularly depression [5] for which antidepressants are commonly prescribed [6,7]. Accordingly, the influence of antidepressants on cancers has become an important research topic.

Notably, inconsistent laboratory and epidemiological findings have been reported for the association between selective serotonin reuptake inhibitors (SSRIs) and HCC. A positive correlation was observed between paroxetine and the increased incidence of liver cancer in male mice [6], as was the administration of sertraline with benign liver tumors [7]. Conversely, fluoxetine was demonstrated to induce apoptotic effects in HCC cell lines [8,9]. Our previous study found that a recipe of tricyclic antidepressants and SSRIs was associated with a lower HCC risk [10,11]. In a nationwide population-based study, we first reported that fluoxetine, sertraline, paroxetine, citalopram, escitalopram, and fluvoxamine were dose-dependently associated with lower HCC risk [11]. Laboratory data have also revealed that the SSRIs fluoxetine and sertraline can induce apoptosis in human hepatoma cells [8,9]. SSRIs, such as citalopram, exert cytotoxic effects on liver cancer cells by causing apoptosis in HepG2 cells, the increased formation of NFκB-dependent reactive oxygen species (ROS), and the increased release of cytochrome c [12].

Escitalopram (Lexapro), an SSRI, is commonly prescribed to patients with major depressive disorder (MDD) and anxiety [13,14]. Evidence has indicated that escitalopram is a superior antidepressant to placebo and other antidepressants, such as citalopram, paroxetine, fluoxetine, sertraline, duloxetine, and sustained-release venlafaxine [15]. According to a literature review, escitalopram exhibits favorable tolerability and generally causes mild and temporary adverse events [15,16,17]. Notably, our recent study demonstrated that escitalopram induces significant apoptotic cascades in U-87MG cells and autophagy in GBM8401 cells, suggesting anti-cancer activity of escitalopram [10]. However, little is known about the effects of escitalopram on HCC.

Evidence has indicated that the p53 protein encoded by the TP53 gene is the most frequently mutated protein during carcinogenesis, accounting for nearly 50% of all human cancers [18,19]. A negative association between p53 mutation and overall survival in post-surgery patients with primary liver cancer was reported in both a cohort study and meta-analysis of 988 patients [20], indicating the importance of p53 mutation in liver cancer. Therefore, HepG2 cells carrying wild-type p53 and Huh-7 cells carrying mutated p53 at codon 220 (A:T-->G:C) [21] were used to investigate the potentials of escitalopram for HCC treatment. Additionally, a nationwide population-based cohort study was also used to investigate the effect of escitalopram on HCC.

## 2. Results

### 2.1. Escitalopram Inhibits the Growth of HepG2 and Huh-7 Cells

MTT was used to examine the viability of HepG2 and Huh-7 in the presence of different concentrations of escitalopram (Figure 1). Significantly lower viability of HepG2 and Huh-7 was detected in the presence of 0.2 mM escitalopram for 24 h as compared to THLE-3 cells (Figure 1A). Compared with THLE-3 cells, the viability of both HepG2 and Huh-7 cells was significantly lower after treatment with 0.1 or 0.2 mM escitalopram for 48 h (Figure 1B). Moreover, escitalopram significantly and dose-dependently increased the levels of p53 and p21 proteins in both HepG2 and Huh-7 cells (Figure 2A–C). Conversely, escitalopram significantly and dose-dependently decreased the levels of cyclin D1 in both HepG2 and Huh-7 cells (Figure 2A–C).

### 2.2. Escitalopram Induces Autophagy in HepG2 and Huh-7 Cells

To investigate whether autophagy is involved in escitalopram-induced cell death, specific antibodies against LC3-II were used to perform the immunofluorescence staining. Apparently higher LC3-II levels were observed in both HepG2 and Huh-7 cells after treatment with 0.05, 0.1 or 0.2 mM escitalopram for 24 h (Figure 3A). Accordingly, a significantly higher ratio of LC3-II/LC3-I was detected in both HepG2 and Huh-7 cells after treatment with 0.05, 0.1 or 0.2 mM escitalopram for 24 h (Figure 3B,C). To further confirm the induction of autophagy by escitalopram, antibodies against ATG-3, ATG-5, ATG-7 and Beclin-1 proteins were used to perform immunoblotting. The expressions of Atg-3, Atg-5, Atg-7 and Beclin-1 proteins were significantly higher in both HepG2 and Huh-7 cells (Figure 4A) after treatment with 0.1 or 0.2 mM escitalopram. Figure 4B,C presents the quantitative levels of Atg-3, Atg-5, Atg-7 and Beclin-1 proteins relative to that of β-actin. Chloroquine (CQ), a validated autophagy inhibitor, was further adopted to confirm the involvement of autophagy to escitalopram-induced death in HepG2 and Huh-7 cells (Figure 5). Pretreatment with CQ blocks lysosomal degradation and prevents LC3-II and p62 break down, leading to the accumulation of LC3-II and p62. After treatment with 25 μM CQ for an hour and subsequent 0.2 mM escitalopram for 24 h, significantly decreased p62 and increased LC3-II levels were observed in both liver cancer cell lines treated with 0.2 mM escitalopram.

### 2.3. Escitalopram Attenuates the Growth of Xenografted Huh-7 Cells in SCID Mice

To verify the effects of escitalopram in vivo, xenografted tumors were generated by subcutaneously injecting 5 × 10^6^ Huh-7 cells into SCID mice. When the tumor volume was approximately 20 mm^3^, the mice were treated daily with PBS (C group), 2.5 mg/kg escitalopram (L group) and 12.5 mg/kg escitalopram (H group) through oral gavage. Although the mean tumor volume in the mice treated with 2.5 mg/kg escitalopram is markedly smaller than that in the mice treated with PBS, no significant difference was found in mean tumor volume between the C and L groups (Figure 6A). Notably, significantly smaller mean tumor volume in the mice treated with 12.5 mg/kg escitalopram was detected versus that in the mice treated with PBS (Figure 6A). Figure 6B shows the representative xenografted tumors retrieved at the end of the experiments.

### 2.4. The Risk of HCC Was Negatively Correlated with Escitalopram Use

The two study groups were patients using Lexapro (n = 167,835) and those who had never used escitalopram (n = 167,835) between 2005 and 2013, who were identified from the NHIRD. Their demographic characteristics, comorbidities, and concomitant medication use are listed in Table 1. No significant differences were noted for sex, age, and urbanization level, but significant differences were noted in related diseases (HBV, HCV, alcohol-related diseases, nonalcoholic steatohepatitis, chronic obstructive pulmonary diseases, diabetes (COPD), liver cirrhosis (LC), chronic kidney disease (CKD), and hypertension (HTN)) and concomitant medication use (angiotensin-converting-enzyme inhibitors (ACEIs), aspirin, metformin, and statin) (*p* < 0.001) between the two groups. Notably, patients who used escitalopram exhibited a significantly decreased risk of liver cancer (*p* < 0.001). Cox proportional hazard regression models were used to examine the risk of liver cancer (Table 2). The risk of liver cancer was positively correlated with age, male sex, presence of related diseases, and concomitant medication use. Univariate and multivariate analyses revealed a significantly lower risk of liver cancer in Lexapro users (univariable analysis: HR: 0.68, 95% CI: 0.59–0.78, *p* < 0.0001; multivariable analysis: HR: 0.55, 95% CI: 0.47–0.63, *p* < 0.0001). Similar results were observed in the analysis of competing risks (univariable analysis: HR: 0.68, 95% CI: 0.59–0.78, *p* < 0.0001; multivariable analysis: HR: 0.56, 95% CI: 0.49–0.65, *p* < 0.0001). Compared with patients who had never used escitalopram, patients who used escitalopram had a significantly lower cumulative incidence of liver cancer during the follow-up periods (log-rank test: *p* < 0.001, Figure 7).

## 3. Discussion

Autophagy is a natural process in eukaryotic cells that maintains homeostasis and ensures cell survival in response to stress conditions, such as nutrition deprivation and hypoxia [22]. Autophagy dysfunction is associated with various diseases, including cancer, and such cell mechanisms have been investigated in pancreatic cancer, renal cell carcinoma, lung cancer, and melanoma [23]. Notably, increased research attention has been paid to autophagy induction in cancer therapy, although some studies suggest that autophagy will promote cancer development, and others suggest that it will inhibit cancer development [24,25]. Although the role of autophagy in cancer development is controversial, various studies have indicated that induction of autophagy may be a new target of cancer therapy [26,27,28]. Accordingly, this study is the first to demonstrate the anti-HCC activity of escitalopram by inducing autophagy in HepG2 and Huh-7 cells.

Studies have provided conflicting results on the effects of SSRIs on tumor growth [6,7,8,9,11,12]. A previous study indicated that use of paroxetine is associated with a substantial increase in breast cancer risk [29]. A recent study also reported that women with breast and ovarian cancer receiving SSRIs exhibit increased recurrence risks and mortality rates [30]. Conversely, some SSRIs can induce apoptosis in Burkitt lymphoma cells by triggering calcium flux, tyrosine phosphorylation and reduction of the c-myc and nm23 genes [31]. Other studies also reported that certain SSRIs induce apoptosis in colorectal cancer cells through upregulation of the mitogen-activated protein kinase (MAPK) cascade [32] and in hepatocellular carcinoma cells by inducing the caspase pathway [33]. Although the underlying mechanisms of the distinct effects of SSRIs are still unclear, recent evidence has indicated that the diverse roles of SSRIs in cancer growth are associated with the patient’s immune function [34].

Evidence has indicated that serotonin (5-HT) has an intensive effect on immune function, which means that SSRIs also have strong influences on the immune system [35]. Notably, different SSRIs exhibit different effects on immune responses. Previous studies have reported that fluvoxamine increases natural killer cells proliferation in cancer patients [36], whereas fluoxetine suppresses immune response [37]. Many studies have indicated that SSRI-influenced immune responses mainly result from direct interaction with immune cells and changes in neurotransmitter concentration [38]. These interactions in tumor microenvironment and secondary lymphoid organs affect cancer pathogenesis [39]. However, more investigations are required to verify the precise mechanism of the distinct roles of SSRIs on immune regulation in HCC therapy.

As reported in a previous study, Huh-7 cells have an amino acid change of cysteine for tyrosine at codon 220 (A:T—G:C) that results in the loss of p53 function [21]. In this study, escitalopram has a positive regulatory effect on expressions of p53 and p21 proteins in both HepG2 and Huh-7 cells. This raises the possibility that escitalopram could induce p21 expression via a p53-independent pathway, resulting in the decline of cyclin D1.

A concern about the doses of escitalopram adopted in this study should be mentioned. The recommended daily dose of escitalopram in adults is about 20 mg/60 kg (0.33 mg/kg) [13,14,15]. When converting a human dose of escitalopram to a mouse dose, the dose will be approximately 2.97 mg/kg and is similar to the dose used in the mice from the low-dose group of this study. Moreover, a recent study has reported that daily administration of 600 mg/60 kg (10 mg/kg) escitalopram exhibits no adverse symptoms [10]. This evidence suggests the repurposing potentials of escitalopram in treatment of HCC. This merits performing pharmacokinetic experiments such as metabolism and elimination of escitalopram in the blood of mice to verify the precise effective dose of escitalopram in treatment of HCC.

One previous case-control study demonstrated that tricyclic antidepressants and SSRIs are associated with a lower HCC incidence [40]. Similarly, fluoxetine, sertraline, paroxetine, citalopram, escitalopram, and fluvoxamine are associated with lower HCC risk in a dose-dependent manner [41]. The present study extended the finding by the design of a cohort study. Another recent study also showed that ATD use, especially a relatively high cumulative dose of SSRIs, in HCV-infected patients was associated with reduced HCC risk [42]. The series of epidemiology studies demonstrate the potential effect of SSRIs on the HCC. Due to the limitation of epidemiology studies on causal inference, such as unmeasured confounding factors, further triangulation studies are warranted [43]. Therefore, the present study provides a new model to replicate the results found from epidemiology design to simultaneous cell study and animal study on the same target drug. 

## 4. Materials and Methods

In this study, cell and animal experiments were first adopted to verify the effects of escitalopram on HCC cells and xenograft tumor. To explore whether the effects of escitalopram are clinically consistent with laboratory findings, a nationwide population-based cohort study was further performed to examine the association between escitalopram and HCC risk.

### 4.1. Cell Culture and Escitalopram

HepG2 cells and THLE-3 cells (normal human liver epithelial cell) were purchased from American Type Culture Collection (ATCC), and Huh-7 cells were purchased from JCRB Cell Bank and were maintained to manufacturers’ instructions (Gibco, Brooklyn, NY, USA; Lonza/Clonetics, Walkersville, MD, USA). Escitalopram oxalate (Escitalopram, Staten Island, NY, USA) was obtained from Chiayi Chang Gung Memorial Hospital.

### 4.2. 3-(4,5-Cimethylthiazol-2-yl)-2,5-Diphenyl Tetrazolium Bromide Assay

Briefly, cells were seeded at a density of 5 × 10^3^ cells/well into a 96-well culture plate and maintained overnight in a cell culture instrument. These cells were treated with different doses of escitalopram for 1 or 2 days. Next, the medium was replaced with 0.2 mL of 3-(4,5-cimethylthiazol-2-yl)-2,5-diphenyl tetrazolium bromide (MTT) reagent (0.5 mg/mL), with cells incubated for another 4 h. The formazan crystals were dissolved in a 0.2 mL dimethyl sulfoxide (DMSO), and the absorbance was detected at 570 nm (SpectraMax M5, San Jose, CA, USA).

### 4.3. Immunofluorescence Staining

After treating with different concentrations of escitalopram for 24 h, the cells on cover-slides were immersed in 4% paraformaldehyde and socked in 0.3% Triton X-100 for 10 min. Coverslips were soaked in blocking solution at 25 °C, reacted with antibodies against LC3-II (Sigma-Aldrich, St. Louis, MO, USA). Subsequently, the coverslips were incubated with Alexa Fluor-conjugated secondary antibodies (Abcam Ltd., Cambridge, UK) for 60 min. After mounting in ProLongTM Gold Antifade Mountant with DAPI (Thermo Fisher Scientific Inc., Waltham, MA, USA) for 10 min, the images were detected using a fluorescence microscope (Carl Zeiss Microscopy, White Plains, NY, USA).

### 4.4. Immunoblotting

The expression of cell cycle- and autophagy-related proteins was detected though immunoblotting. Briefly, antibodies against p53 (PharMingen, San Diego, CA, USA), p21 (PharMingen), cyclin D1 (SANTA CRUZ, Starr County, TX, USA), LC3 (Novus Biologicals, Minneapolis, MN, USA), Atg-3 (Sigma-Aldrich), Atg-5 (Sigma-Aldrich), Atg-7 (Sigma-Aldrich), Beclin-1 (Novus Biologicals), and β-actin (Upstate, VA, USA) were diluted 1000 times in PBS with 2.5% bovine serum albumin (BSA), and cells were incubated with these antibodies for 3 h at room temperature with gentle agitation at 25 °C. Subsequently, cells were incubated with secondary antibody conjugated with horseradish peroxidase (HRP) for another hour. Cells were washed with PBS-Tween; then, the HRP Chemiluminescent Substrate (Millipore, Burlington, MA, USA) was used to detect antigen–antibody (Ag–Ab) complexes. The density of Ag–Ab complexes was then quantified by a densitometry device (Appraise, Overland Park, KS, USA).

### 4.5. Xenograft Study

Teen male SCID mice (5-week old) were purchased from the National Center for Experimental Animals of Taiwan and were bred in a specific-pathogen-free (SPF) device with 12 h light/12 h dark cycle. All study protocols were approved by the Institutional Animal Care and Use Committee of Chiayi Chang Gung Memorial Hospital, Taiwan (approval number: 2015040102). The study was carried out in compliance with the ARRIVE guidelines (Animal Research: Reporting of In Vivo Experiments). After a week, a total of 5 × 10^6^ Huh-7 cells were hypodermic injected into the flank of mice. While tumor volume reached almost 20 mm^3^, the mice were randomly divided into two groups as control (C), low-dose (L) and high-dose (H) groups. The mice from C, L and H groups were orally administered daily with PBS, 2.5 and 12.5 mg/kg escitalopram, respectively. Tumor diameters and volume were measured and calculated weekly using a caliper. The mice were sacrificed after 4 weeks of treatment, and the tumors were excised and weighed.

### 4.6. Nationwide Population-Based Cohort Study

The Institutional Review Board (IRB) of Chang Gung Medical Foundation approved this study (IRB certificate approval number: 201502598B0D001, 7 May 2015). The National Health Insurance (NHI) program was launched in Taiwan on 1 March 1995. Almost all residents of Taiwan are covered by the NHI program. The detailed medical information of insurants covered by this program, which includes demographic data and records of outpatient visits, hospital admissions, diagnosis, prescriptions and dosages, and medical procedures, are collated in the National Health Insurance Research Database (NHIRD). To investigate the association between HCC risk and escitalopram, we conducted a nationwide population-based retrospective cohort study. We analyzed the data from the NHIRD for the period between 1 January 2005 and 31 December 2013. HCC patients were identified as those having a record of two or more outpatient visits or at least one inpatient visit for HCC diagnosis. HCC diagnosis was based on the International Classification of Diseases, Ninth Revision, Clinical Modification (ICD-9-CM) code of 155.xx. In addition, we linked the data to the Catastrophic Illness Database to verify the diagnosis. In Taiwan, the Registry of the Catastrophic Illness Database is used for patients with definitive cancer diagnoses. HCC diagnosis was made by a physician and was reviewed by a panel of NHI. Moreover, the diagnosis must be confirmed based on tissue pathology. We identified medications using the Anatomical Therapeutic Chemical (ATC) classification system. The ATC code for escitalopram is N06AB10. Participants were divided into two groups: escitalopram users and nonusers between 2005 and 2013. The study outcome was the date of the first HCC diagnosis. The exclusion criteria for escitalopram users were as follows: patients with cancer diagnosis before the use of escitalopram or during the first 6 months of escitalopram use and patients using escitalopram for less than 28 days.

### 4.7. Statistics

GraphPad Prism 5 software (GraphPad Software, La Jolla, CA, USA) was adopted for bench work analysis. Specifically, one-way analysis of variance (one-way ANOVA) and Tukey’s multiple comparisons test were performed to determine the significance of the results. Data are represented as mean ± SEM. Three repeated experiments were performed. We used SAS version 9.4 (SAS Institute, Cary, NC, USA) for epidemiological analysis. The covariates included sociodemographic factors, comorbidities, and concomitant medications. First, we conducted descriptive analysis for categorical variables and continuous variables for analyzing demographic factors, comorbidities, and concomitant medication use in Lexapro users and nonusers. Furthermore, we used Cox proportional hazard regression models (univariable and multivariable models) to evaluate the risk of HCC. A *p* value < 0.05 was considered statistically significant.

## 5. Conclusions

Repurposing of old drugs for different diseases, especially cancers, has various advantages such as lower risk process and cost and time effectiveness [44]. Clinical evidence has revealed that escitalopram has favorable tolerability and generally mild and temporary adverse events relative to other SSRIs [15,17,45]. In the current study, laboratory and nationwide population-based cohort evidence demonstrated the attenuated effects of escitalopram on HCC. Although these findings suggest potential roles of escitalopram on HCC treatment, further replication studies and clinical trials are warranted to prove the clinical use of escitalopram on HCC.

## Figures and Tables

**Figure 1 ijms-23-09247-f001:**
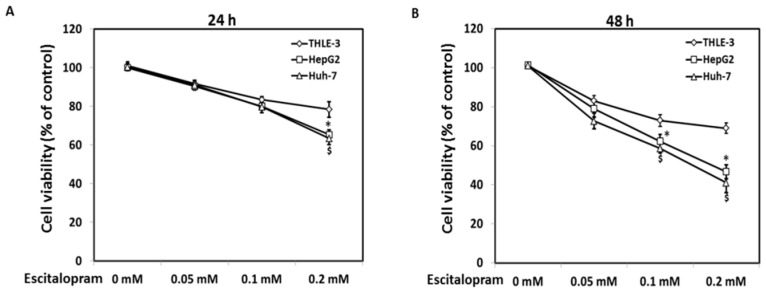
Effects of escitalopram on cell viability of THLE-3, HepG2 and Huh-7 cells. Survival ratios of THLE-3, HepG2, and Huh-7 cells after treatment with different concentrations of escitalopram for (**A**) 24 or (**B**) 48 h. Similar results were obtained in three repeated experiments. The symbols * and $ indicate a significant difference; *p* < 0.05, relative to controls (0 mM) and THLE-3 cells.

**Figure 2 ijms-23-09247-f002:**
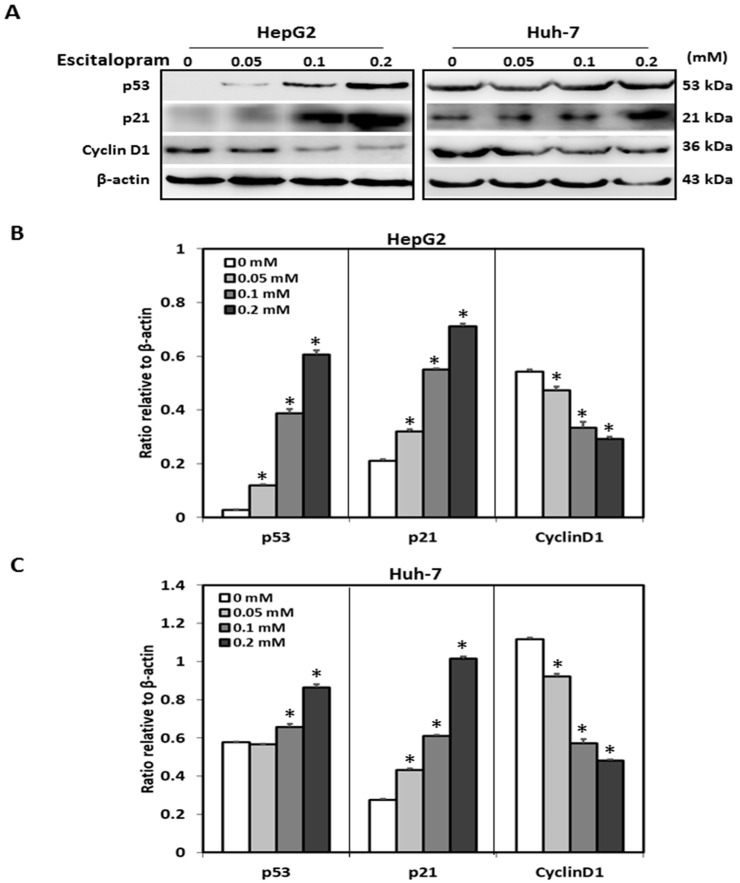
Effects of escitalopram on the expressions of cell cycle-related proteins in THLE-3, HepG2 and Huh-7 cells. (**A**) Expressions of p53, p21 and cyclin D1 proteins in HepG2 and Huh-7 cells treated with different concentrations of escitalopram for 24 h. Bars represent the levels of p53, p21 and cyclin D1 proteins on the basis of β-actin in (**B**) HepG2 and (**C**) Huh-7cells. Similar results were obtained in three repeated experiments. The symbol * indicates a significant difference; *p* < 0.05, relative to controls (0 mM).

**Figure 3 ijms-23-09247-f003:**
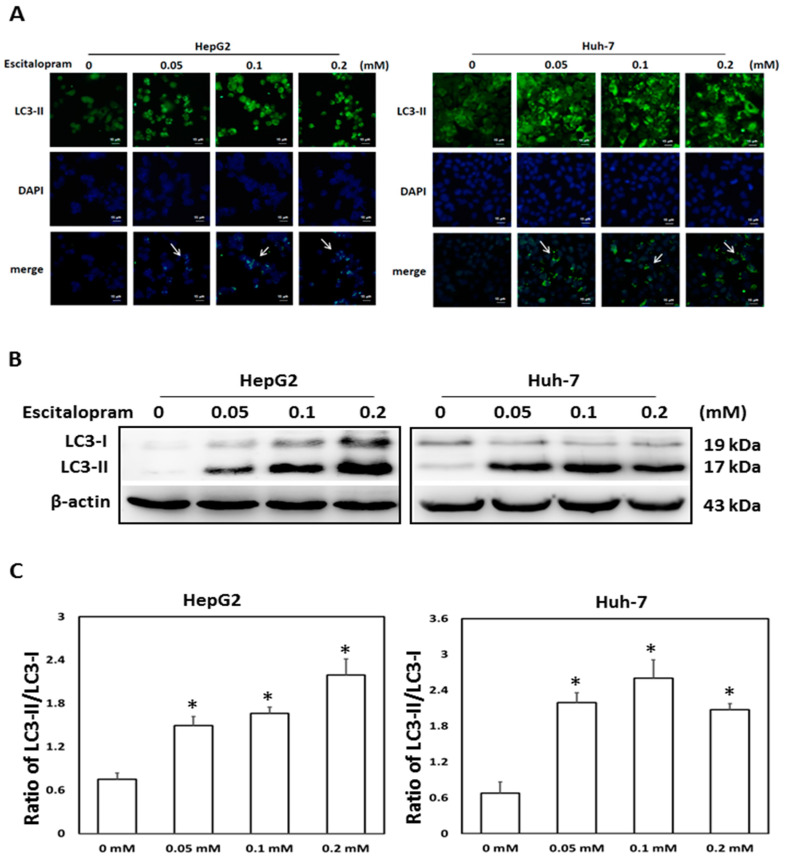
Expression of LC3-II proteins in HepG2 and Huh-7 cells treated with escitalopram. (**A**) The images of immunofluorescence staining with antibodies against LC3-II and DAPI as well as the merged images of DAPI and LC3-II in HepG2 and Huh-7 cells treated with different concentrations of escitalopram for 24 h. (**B**) Expressions of LC3-I and LC3-II proteins in HepG2 and Huh-7 cells treated with different concentrations of escitalopram for 24 h. (**C**) Bars represent the ratios of LC3-II/LC3-I. Similar results were obtained in three repeated experiments. The symbol * indicates a significant difference relative to the control group (0 mM); *p* < 0.05.

**Figure 4 ijms-23-09247-f004:**
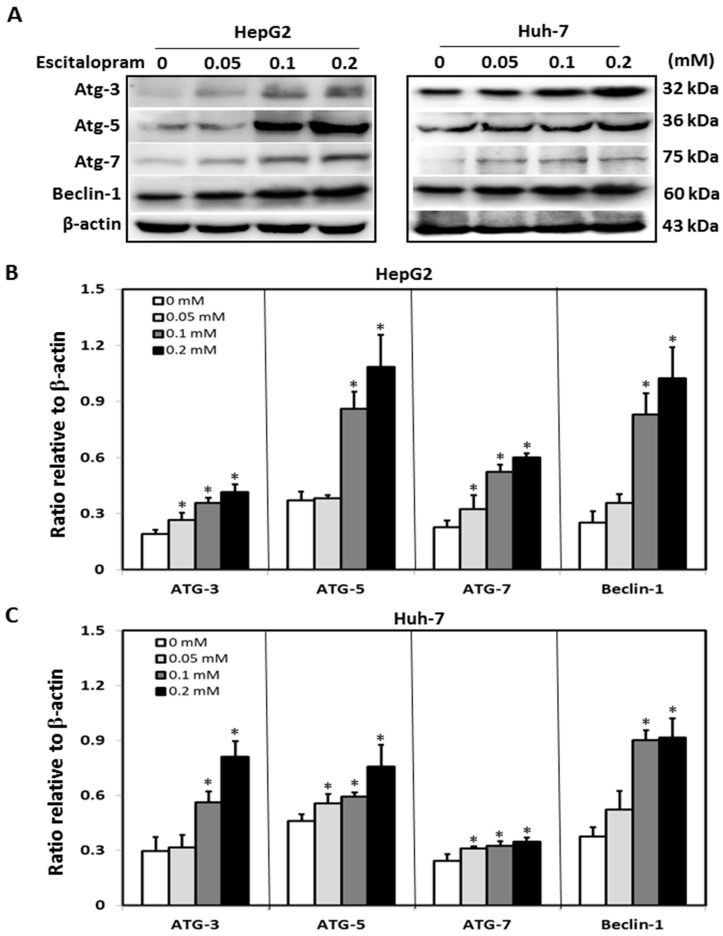
Expressions of the autophagy-related proteins in HepG2 and Huh-7 cells treated with escitalopram. (**A**) Expressions of Atg-3, Atg-5, Atg-7 and Beclin-1 proteins in HepG2 and Huh-7 cells after treatment with different concentrations of escitalopram for 24 h. Bars represent the levels of Atg-3, Atg-5, Atg-7 and Beclin-1 proteins on the basis of β-actin in (**B**) HepG2 and (**C**) Huh-7 cells. Similar results were obtained in three repeated experiments. The symbol * indicates a significant difference relative to the control group; *p* < 0.05.

**Figure 5 ijms-23-09247-f005:**
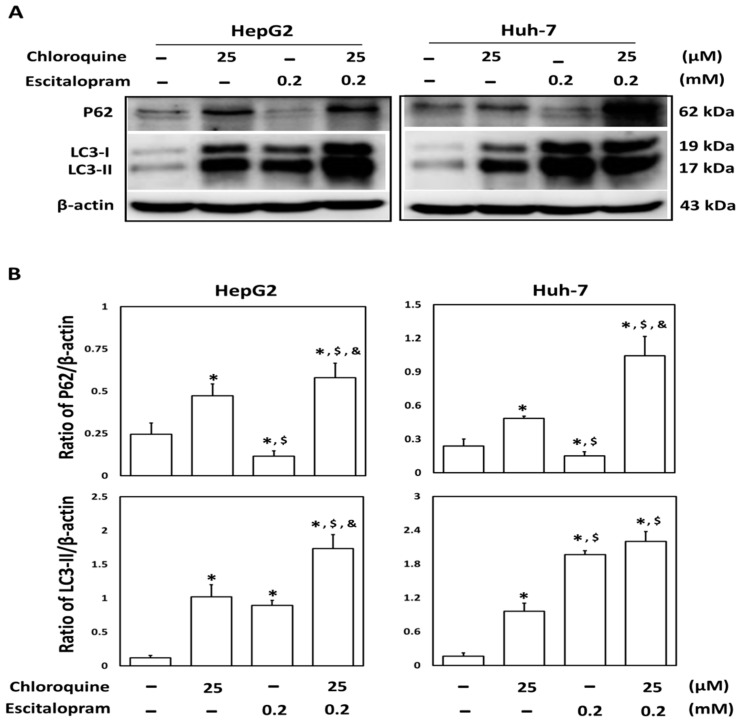
Involvement of autophagy in the response of liver cancer cells treated with escitalopram. HepG2 and Huh-7 cells were pretreated with 25 μM chloroquine for an hour before escitalopram treatment (0.2 mM) for 24 h. (**A**) Expressions of p62 and LC3-II proteins in HepG2 and Huh-7 cells. (**B**) Bars represent the ratios of p62/β-actin and LC3-II/β-actin. Similar results were obtained in three repeated experiments. The symbols *, $ and & indicate a significant difference (*p* < 0.05) compared with control, chloroquine and escitalopram, respectively.

**Figure 6 ijms-23-09247-f006:**
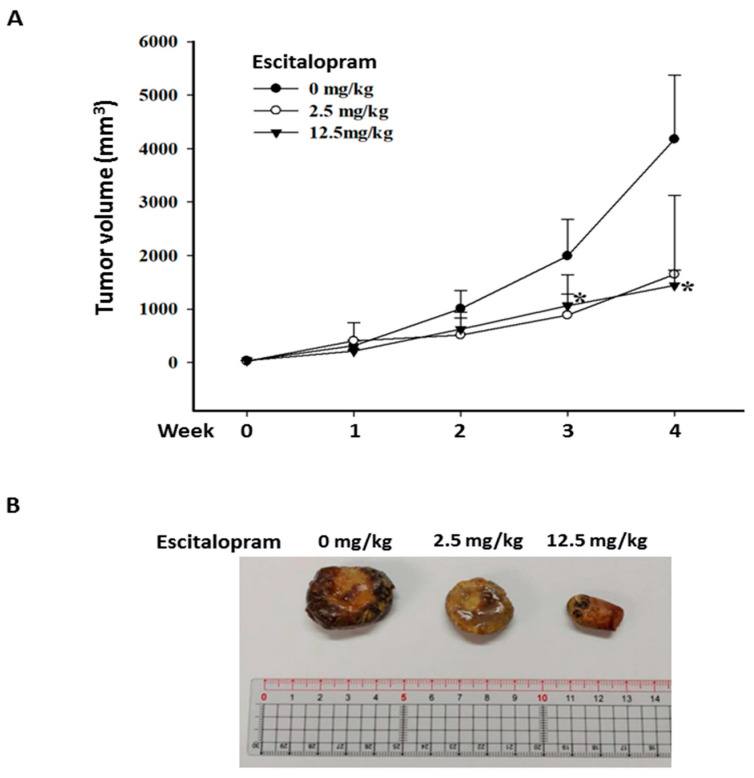
Escitalopram inhibits xenografted Huh-7 cells in SCID mice. (**A**) The different groups of mice were administered PBS (control group), 2.5 mg/kg (low dose group) and 12.5 mg/kg (high dose group) escitalopram daily through oral gavage for 28 days. (**B**) Representative images of the excised xenograft tumors from the different groups of mice. The symbol * indicates a significant difference relative to the control group; *p* < 0.05.

**Figure 7 ijms-23-09247-f007:**
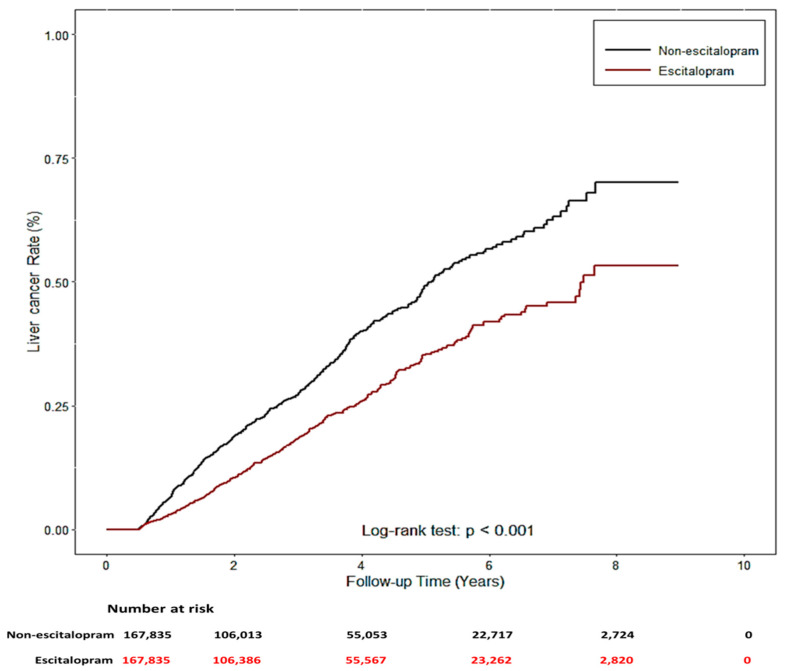
The cumulative incidence of liver cancer with/without escitalopram used.

**Table 1 ijms-23-09247-t001:** Demographic characteristics of the participants in the population-based study.

	Use of Escitalopram n = 167,835	Never Used Escitalopram n = 167,835	*p* Value
**Age, N (%)**			
<50	88,645 (52.8%)	88,645 (52.8%)	1.0000
50~65	47,015 (28.0%)	47,015 (28.0%)	
≥65	32,175 (19.2%)	32,175 (19.2%)	
**Sex, N (%)**			
Female	106,034 (63.2%)	106,034 (63.2%)	1.0000
Male	61,801 (36.8%)	61,801 (36.8%)	
**Urbanized level of residence**			
1 (City)	45,535 (27.1%)	45,535 (27.1%)	1.0000
2	68,450 (40.8%)	68,450 (40.8%)	
3	20,514 (12.2%)	20,514 (12.2%)	
4 (Villages)	33,336 (19.9%)	33,336 (19.9%)	
**Income**			
Income = 0	48,890 (29.1%)	48,890 (29.1%)	1.0000
1 <= income <= 15,840	32,163 (19.2%)	32,163 (19.2%)	
15,841 <= income <= 25,000	61,756 (36.8%)	61,756 (36.8%)	
Income >= 25,001	25,026 (14.9%)	25,026 (14.9%)	
**Related diseases, N (%)**			
HBV	10,118 (6.0%)	6108 (3.6%)	<0.001
HCV	5122 (3.1%)	2694 (1.6%)	<0.001
Alcohol-related disease	8378 (5.0%)	2716 (1.6%)	<0.001
Non-alcoholic steatohepatitis	10,862 (6.5%)	6041 (3.6%)	<0.001
COPD	24,128 (14.4%)	15,541 (9.3%)	<0.001
Diabetes	29,208 (17.4%)	21,737 (13.0%)	<0.001
LC	4006 (2.4%)	2394 (1.4%)	<0.0001
CKD	4588 (2.7%)	3363 (2.0%)	<0.001
HTN	59,358 (35.4%)	44,710 (26.6%)	<0.001
**Drug use ^a^, N (%)**			
ACEIs	33,885 (20.2%)	26,955 (16.1%)	<0.0001
Aspirin	47,484 (28.3%)	32,957 (19.6%)	<0.001
Metformin	20,182 (12.0%)	16,931 (10.1%)	<0.001
Statin	37,582 (22.4%)	26,280 (15.7%)	<0.001
**Outcome, N (%)**			
Liver cancer	333 (0.2%)	489 (0.3%)	<0.001

Abbreviations: HBV, hepatitis B virus; HCV, hepatitis C virus; COPD, chronic obstructive pulmonary disease; LC, liver cirrhosis; CKD, chronic kidney disease; HTN, hypertension; ACEIs, angiotensin-converting-enzyme inhibitors. ^a^ Drug use was defined as the frequency in years prior to endpoint.

**Table 2 ijms-23-09247-t002:** Cox proportional hazards regression model analysis for risk of liver cancer.

	Univariable		Multivariable	
Variables	HR (95% CI)	*p*-Value	HR (95% CI)	*p* Value
Escitalopram	0.68 (0.59–0.78)	<0.0001	0.55 (0.47–0.63)	<0.0001
Age 50~65	5.61 (4.49–7.01)	<0.0001	4.24 (3.35–5.35)	<0.0001
Age ≥ 65	11.92 (9.63–14.74)	<0.0001	8.11 (6.34–10.37)	<0.0001
Male sex	2.16 (1.89–2.48)	<0.0001	2.01 (1.73–2.32)	<0.0001
Urbanized level of residence 1	0.68 (0.56–0.83)	0.0001	1.12 (0.92–1.38)	0.2647
Urbanized level of residence 2	0.77 (0.65–0.91)	0.0024	1.08 (0.90–1.28)	0.4097
Urbanized level of residence 3	1.01 (0.81–1.26)	0.9546	1.06 (0.85–1.33)	0.5971
1 <= income <= 15,840	1.33 (1.05–1.69)	0.0204	0.88 (0.69–1.12)	0.2988
15,841 <= income <= 25,000	2.26 (1.86–2.73)	<0.0001	1.27 (1.04–1.55)	0.0202
Income >= 25,001	1.55 (1.22–1.98)	0.0004	1.00 (0.77–1.30)	0.9981
**Related diseases**				
HBV	6.77 (5.76–7.94)	<0.0001	3.07 (2.58–3.66)	<0.0001
HCV	16.55 (14.17–19.33)	<0.0001	4.18 (3.47–5.02)	<0.0001
Alcohol-related disease	3.22 (2.56–4.06)	<0.0001	1.15 (0.89–1.49)	0.2732
Non-alcoholic steatohepatitis	4.40 (3.68–5.25)	<0.0001	1.40 (1.15–1.69)	0.0006
COPD	2.33 (1.98–2.74)	<0.0001	0.97 (0.82–1.15)	0.7186
Diabetes	3.58 (3.11–4.13)	<0.0001	1.30 (1.06–1.59)	0.0107
LC	24.91 (21.47–28.90)	<0.0001	6.68 (5.54–8.06)	<0.0001
CKD	2.68 (1.96–3.65)	<0.0001	0.87 (0.63–1.19)	0.3765
HTN	3.60 (3.13–4.14)	<0.0001	1.27 (1.05–1.52)	0.0122
**Drug use**				
ACEI	3.00 (2.61–3.45)	<0.0001	1.10 (0.92–1.31)	0.3083
Aspirin	2.58 (2.25–2.96)	<0.0001	0.91 (0.77–1.07)	0.234
Metformin	3.11 (2.67–3.61)	<0.0001	1.30 (1.05–1.61)	0.0166
Statin	1.05 (0.89–1.25)	0.5529	0.53 (0.44–0.64)	<0.0001

Abbreviations: HBV, hepatitis B virus; HCV, hepatitis C virus; COPD, chronic obstructive pulmonary disease; LC, liver cirrhosis; CKD, chronic kidney disease; HTN, hypertension; ACEIs, angiotensin-converting-enzyme inhibitors.

## Data Availability

The data presented in this study are available on request from the corresponding author.

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
