# Peer review of "Protective Effect of Escitalopram on Hepatocellular Carcinoma by Inducing Autophagy"

_ijms, 2022, doi:10.3390/ijms23169247_

Round 1

Reviewer 1 Report

Although this is an interesting topic, the introduction and the literature review is poorly presented. An additional point is the methodology section. It is not clear what happened, how it was assessed, therefore this section must be described in more detail. The results and conclusions should be more detailed and explain to the reader the clinical implications.

Reviewer 2 Report

- Please indicate the real [escitalopram] found in blood (introduction paragraph). Is it in the range studied? 

- Please indicare in 2.3 the [escitalopram] found in mice blood. 

- Surprise that  in "Demographic characteristics of the participants in the population-based study - Table 1" the number of participants in each group is the same and also in the range of ages!

- "Overexpression of 5-hydroxytryptamine (5-HT) in human cancer promotes colorectal tumor invasion by activating the Axin1/β-catenin/MMP-7 pathway [25]. Conversely, the antitumor effects of SSRIs on various types of cancer cells have also been reported." Sounds a bite contradictory if it is not explained. 

- Conclusion should mention the possibility cited that ". Conversely, if the immune system is functional, treatment with SSRIs adversely affects cancer patients"

Reviewer 3 Report

The manuscript by Chen and coworkers deals with the putative mechanisms underlying the anticancer activity of escitalopram, an antidepressant commonly prescribed to cancer patients. The authors demonstrate that escitalopram reduces the viability and triggers autophagy in hepatoma cells, and significantly attenuates the onset of HCC in a cohort of escitalopram-treated patients compared to non-treated controls. The authors suggest that escitalopram-induced autophagy might account for the reduced HCC incidence.

The research is interesting, in particular due to the fact that the prognosis of HCC patients is normally poor. In spite of this fact, in my opinion, the manuscript has some drawbacks that have been detailed below to the attention of the authors.

Major criticisms

Fig. 1B. The authors state: ‘…Compared with THLE-3 cells, the viability of both Huh-7 and HepG2 cells was significantly higher after treatment with 0.1 mM or 0.2 mM escitalopram for 48 h (Fig. 1B).’ However, in fig. 1b the viability of Huh7 and HEPG2 cells is lower than that of THLE-3 cells. Unless I miss some particular, this sentence seems incorrect and needs revision.

Results, Paragraph 2.2. A) The authors state that escitalopram triggers autophagy. However, the increase in the ratio of LC3-II/LC3-I, as used in Figure 3B, should be verified also in the presence of validated autophagy inhibitors. As widely reported, an isolate increase of LC3-II could also be achieved by a blockade, rather than an increase, of autophagy (on this, please see Mizushima & Yoshimori, doi: 10.4161/auto.4600). Since the induction of autophagy could be a key mechanism of the antitumor action of escitalopram, autophagy induction must be demonstrated by strictly complying with the state-of-the-art autophagy detection guidelines.

B) Still on this, the authors should use validated autophagy inhibitors to definitely support the role of autophagy-induced cell death in the escitalopram-induced viability reduction observed in the hepatoma cell lines used for this research.

Results, Paragraph 2.4. Please, confirm that the patient groups analyzed in this study were composed exactly of the same number of subjects (167835, as reported in the text).

Minor criticisms.

The style of the manuscript would require attention. As a single example, please note that, in the first paragraph of the Results, there are three almost identical sentences […Compared with THLE-3 cells, the viability of Huh-7 cells was significantly lower after treatment with 0.1 mM escitalopram for 24 h (Fig. 1A). Compared with THLE-3 cells, the viability of Huh-7 and HepG2 cells was significantly lower after treatment with 0.2 mM escitalopram for 24 h. Compared with THLE-3 cells, the viability of both Huh-7 and HepG2 cells was significantly higher after treatment with 0.1 mM or 0.2 mM escitalopram for 48 h (Fig. 1B)]. If possible, these parts should be revised.

Round 2

Reviewer 3 Report

I have no other requests for the Authors.

Author Response

Thanks for your kind comments.